# Natural Lipid Extracts as an Artificial Membrane for Drug Permeability Assay: In Vitro and In Silico Characterization

**DOI:** 10.3390/pharmaceutics15030899

**Published:** 2023-03-10

**Authors:** Anna Vincze, Gergely Dékány, Richárd Bicsak, András Formanek, Yves Moreau, Gábor Koplányi, Gergely Takács, Gábor Katona, Diána Balogh-Weiser, Ádám Arany, György T. Balogh

**Affiliations:** 1Department of Chemical and Environmental Process Engineering, Budapest University of Technology and Economics, Műegyetem rkp. 3, H-1111 Budapest, Hungary; 2ESAT-STADIUS KU LEUVEN, 3001 Leuven, Belgium; 3Department of Measurement and Information Systems, Faculty of Electrical Engineering and Informatics, Budapest University of Technology and Economics, Műegyetem rkp. 3, H-1111 Budapest, Hungary; 4Department of Organic Chemistry and Technology, Faculty of Chemical Technology and Biotechnology, Budapest University of Technology and Economics, Műegyetem rkp. 3, H-1111 Budapest, Hungary; 5Mcule.com Kft, Bartók Béla út 105-113, H-1115 Budapest, Hungary; 6Institute of Pharmaceutical Technology and Regulatory Affairs, Faculty of Pharmacy, University of Szeged, Eötvös Str. 6, H-6720 Szeged, Hungary; 7Department of Physical Chemistry and Materials Science, Faculty of Chemical Technology and Biotechnology, Budapest University of Technology and Economics, Műegyetem rkp. 3, H-1111 Budapest, Hungary; 8Institute of Pharmacodynamics and Biopharmacy, Faculty of Pharmacy, University of Szeged, Eötvös u. 6, H-6720 Szeged, Hungary

**Keywords:** natural lipid extract, total and polar lipids, brain–heart–liver lipids, tissue-specific permeability, PAMPA, physicochemical and in silico characterization

## Abstract

In vitro non-cellular permeability models such as the parallel artificial membrane permeability assay (PAMPA) are widely applied tools for early-phase drug candidate screening. In addition to the commonly used porcine brain polar lipid extract for modeling the blood–brain barrier’s permeability, the total and polar fractions of bovine heart and liver lipid extracts were investigated in the PAMPA model by measuring the permeability of 32 diverse drugs. The zeta potential of the lipid extracts and the net charge of their glycerophospholipid components were also determined. Physicochemical parameters of the 32 compounds were calculated using three independent forms of software (Marvin Sketch, RDKit, and ACD/Percepta). The relationship between the lipid-specific permeabilities and the physicochemical descriptors of the compounds was investigated using linear correlation, Spearman correlation, and PCA analysis. While the results showed only subtle differences between total and polar lipids, permeability through liver lipids highly differed from that of the heart or brain lipid-based models. Correlations between the in silico descriptors (e.g., number of amide bonds, heteroatoms, and aromatic heterocycles, accessible surface area, and H-bond acceptor–donor balance) of drug molecules and permeability values were also found, which provides support for understanding tissue-specific permeability.

## 1. Introduction

The parallel artificial membrane permeability model (PAMPA) was introduced by Kansy et al. in 1998 [1], and it is the most widely used in vitro, high-throughput system for the investigation of the passive transport processes of drugs without the use of cells. The sandwich-type set-up of two pairs (donor and acceptor side) of 96-well plates allows for a wide range of parameters (pH_donor/acceptor_: 2.0–7.4, T: 10–45 °C, t: 15 min–16 h, selectable plasma proteins and solubility-improving excipients, etc.) and the compatibility of plate-based analytical systems, thus PAMPA models have become increasing popularity. In addition, one of the unique features of PAMPA is that different lipid solutions can act as artificial membranes and they can be robustly created on the hydrophobic filter material of the donor wells. The specificity of the tissue and entry gate for drug absorption of the developed PAMPA models closely relates to the design of its artificial membrane system. Thus, the barrier- or tissue-mimetic properties of PAMPA can be achieved either by using simple hydrophobic solvents (n-hexadecane–*gastrointestinal* (*GI*) [2]; n-dodecane–*buccal* [3]), mono-phospholipids (lecithin/n-dodecane–*GI* [1], phosphatidylcholines (PC)/n-dodecane:n-hexane:chloroform–*corneal* [4]), phospholipid mixtures (PC/phosphatidylserine (PS)/n-dodecane–*blood–brain barrier (BBB)* [5,6]), phospholipids/cholesterol (PC/PS/phosphatidylethanolamine (PE)/phosphatidylinositol (PI)/cholesterol (CHO)/1,7-octadiene–*intestinal brush border membrane* [7]), tissue-specific lipid extracts (brain polar lipid extract (BPLE)/n-dodecane:(n-hexane)–*BBB* [4,8,9], BPLE, heart and liver polar lipid extracts (HPLE, LPLE)/n-dodecane–*tissue-specific phospholipidosis models* [10]) dissolved in a hydrophobic solvent, or synthetic lipids dissolved in a hydrophilic solvent (certramide/CHO/stearic acid/silicon oil–skin [11]). In addition, a very important observation confirmed by Carrara et al. [12] is that the amount of the organic solvent media and the concentration of used phospholipid or lipid mixture play a significant role in the tissue-specific behavior of the artificial membrane. Based on this observation, our group fine-tuned the BBB-specific PAMPA model, which proved that by reducing the amount of n-dodecane in the solvent medium used in addition to BPLE by increasing the proportion of n-hexane (as a volatile co-solvent component) significantly increases the biomimetic nature or *lipid-driven mechanisms* of PAMPA systems [5] (Figure 1).

Due to the needs of the pharmaceutical industry, the development of PAMPA models has focused primarily on modeling the oral absorption of drug candidates and their delivery to the central nervous system (PAMPA-GI and PAMPA-BBB). However, creating new pharmacokinetically relevant PAMPA models is a difficult task, since there is little or no in vivo data available for other target barriers due to technical limitations. A typical example of this is topical drug administration (e.g., trans-dermal or trans-corneal absorption), as it would be difficult to collect in vivo human PK data from both a technical and medical ethical point of view. Likewise, direct in vivo human sample isolation related to individual target tissues (e.g., lymph, muscle, or connective tissue) and organs (e.g., heart, lung, liver, and kidney) for studying systemic drug distribution poses a great challenge. Perhaps this is the reason why neither in the general in vitro cell-free diffusion systems nor in the PAMPA models are there any examples of the use of tissue-specific lipids as artificial membranes or a detailed comparative study. Only six standardized isolates of mammalian tissue-specific lipids can be obtained from commercial sources, sold as total and polar lipid extracts from a bovine heart (HTLE and HPLE) and liver (LTLE and LPLE), as well as a porcine brain (BTLE and BPLE). Polar lipid extracts are derived from the total lipid extract by precipitation with acetone followed by extraction of the acetone-insoluble material with a diethyl ether based on the methods published by Folch et al. [14]. Polar extracts are enriched in polar lipids such as glycerophospholipids, sphingolipids, and glycolipids, while non-polar lipid components discarded during the precipitation process include triglycerides, cholesterol, pigments, etc. [15].

To acquire further knowledge about tissue-specific drug absorption, the aim of the present study is the investigation of the commercially available standardized total and polar lipid extracts in the PAMPA system by applying in silico and experimental parameters. This is the first time that a comprehensive investigation of mammalian-tissue-specific lipids has been conducted. As part of the study, the BPLE of porcine origin was also investigated, which is widely used for modeling penetration through the blood–brain barrier. Based on the composition of the 3 × 2 (brain/heart/liver × total/polar) lipid extracts given by the manufacturer’s certification, statistical and cheminformatic tools were used to characterize the phospholipid composition of the extracts and their net charges based on the ionization state of individual phospholipid molecules formed in the pH 7.4 test medium. The surface charge of the lipid extracts was characterized based on the zeta potential values of the liposomal systems. The extent of and differences in penetration for the six natural lipid-extract-based PAMPA systems were investigated on 32 physicochemically and structurally diverse active agents. Just like natural lipids characterization, the PAMPA permeability values of the investigated compounds were subjected to a statistical analysis using their net charges and medchem-relevant in silico physicochemical and structural descriptors. Our goal is to identify the differences in the interaction of individual active ingredients and natural lipids via permeability measurements. These findings can serve as evidence for the tissue-specific behavior of natural lipids and demonstrate the difference between total and polar extracts. Although extensive PAMPA—in vivo human logBB correlations are available only in the case of BPLE, it is hoped that our study will also provide information on the applicability of natural lipid extracts from the heart and liver in the design of in vitro cell-free permeability tests, as well as the integrability of these models in early-stage drug discovery.

## 2. Materials and Methods

### 2.1. Materials

Lipid extracts of bovine origin (heart polar—HPLE, heart total—HTLE, liver polar—LPLE, and liver total—LTLE) and of porcine origin (brain polar—BPLE and brain total—BTLE) were purchased from Avanti Polar Lipids Inc. (Alabaster, AL, USA). Analytical-grade solvents such as acetonitrile (AcN), formic acid (99%), chloroform, hexane, and dodecane were purchased from Merck KGaA (Darmstadt, Germany). The powder required for the preparation of phosphate-buffered saline (PBS, 0.01 M) solution was also purchased from Merck KGaA (Darmstadt, Germany). The water for buffer and mobile phase preparation was provided by a Millipore MilliQ water purification system. Each active pharmaceutical ingredient (API—see Figure 2) used in the experiment was purchased from Sigma Aldrich (Merck KGaA, Darmstadt, Germany) except for the ‘PGY0072′ compound (the structure can be found in Appendix A), which was provided by Egis Pharmaceuticals Plc (Budapest, Hungary).

### 2.2. Compound Set Selection and Cheminformatic Tools for Physicochemical and Structural Characterization and Structure–Property Interpretation

First, in-stock compounds were screened to meet the following two in-house criteria: (a) medchem: clog*P* < 4, clog*D*_pH7.4_ < 3, TPSA > 50 and (b) analytical: having at least one unsaturated bond or aromatic ring as a chromophore. In the next step, the remaining 172 compound structures were desalted (inactive counter ions eliminated from the structure), and their all-to-all pairwise structural similarity was computed using the Tanimoto similarity metric applied on three different chemical fingerprints: ECFP (radius = 2), FCFP (radius = 2), and MACCS keys, using an RDKit [16]. The molecules were selected manually such that no high similarity (top 2–5 percent of pairwise similarity excluded) between the structures could be observed according to any of the three fingerprints, while also considering analytical feasibility.

The predicted physicochemical parameters used for chemical diversity interpretation were generated by Marvin Sketch, Version: 20.21.0 (Chemaxon Ltd., Budapest, Hungary), which is freely accessible with an academic license. For the structural characterization and visualization of our compound set, we used t-SNE [17] for dimensionality reduction and clustering. The t-SNE abbreviation stands for t-distributed stochastic neighbor embedding. It is an unsupervised, non-linear algorithm that uses and essentially optimizes probability distributions in both high- and low-dimensional space. The t-SNE calculation and the following visualization were carried out using Chemplot library version 1.2.1. [18]. The compounds were described with their 1,024-bit ECFP4 fingerprint as a structural descriptor. Additional physicochemical and specific pharmacokinetic descriptors for the selected molecules used in the cheminformatic analysis were calculated using the freely accessible RDKit v. 2022.03.5 python library and the ACD/Percepta, Version: v2020 Build 3382 (ACD/Labs, Toronto, Ontario, Canada, https://www.acdlabs.com/products/percepta/, accessed on 18 June 2020). PCA analysis and correlation analysis were carried out using the scikit-learn version 1.0.2. python library.

### 2.3. Preparation of Liposomes

Briefly, 5-5 mg of lipids were dissolved in 2 mL chloroform in round-bottom flasks. Thin films were obtained by evaporation of the organic solvent using rotary vacuum evaporation (Büchi R-210, Büchi, Flawil, Switzerland) at 25 °C and 50 rpm under 300 mbar pressure. Thereafter, the lipid films were hydrated with 5 mL PBS (pH 7.4) resulting in liposomes with a lipid concentration of 1 mg/mL. The samples were vortexed at 500 rpm (Biobase MX-S, Jinan, Shandong, China) for 5 min and subsequently, the dispersions were sonicated using an ultrasonic bath (Elmasonic S30, Elma Ultrasonic Technology, Singen, Germany). The liposomes were kept at 4 °C until they were used for further investigation.

### 2.4. Zeta Potential Measurement

The zeta potential of the liposomes was measured using a Malvern Panalitycal ZetaSizer ProBlue instrument equipped with a 633 nm red laser at a detection angle of 173° (back-scattering mode) (Malvern Panalytical Ltd., Malvern, UK). Measurements of the liposome suspensions were performed in 1 × 1 cm cuvettes (DTS0012) in triplicate by applying a DIP cell (ZEN1002) probe at 37.0 °C. For data analysis, ZS Explorer software-2.31 (2022) was used.

### 2.5. PAMPA Measurements

For in vitro PAMPA permeability assays, 16 mg natural polar or total lipid extracts were weighted and dissolved in 600 μL solvent mixture of chloroform, hexane, and dodecane (5:25:70 *v*/*v*%) at 0 °C. Each well of the donor plate (Multiscreen IP Filter plate, 0.54 μm, Millipore, Merck KGaA, Darmstadt, Germany) was coated with 5 μL lipid solution, and after evaporation of the chloroform:hexane solvent mixture, the donor plate was put into the acceptor plate (MSSACCEPT0R, resuable PTFE plate, Millipore, Merck KGaA, Darmstadt, Germany) already filled with 300 μL PBS (pH 7.4) in each well. After that, 150 μL of API solution with a nominal concentration of 100 μM (dissolved in PBS, pH 7.4) was added to each well of the donor plate, in triplicate. The sandwich system was covered with wet paper tissue and a plate lid and incubated for 4 h at 37 °C (Titramax 1000, Heidolph Instruments GmbH & CO. KG, Swabach, Germany). After the plates were disassembled, samples were taken from both the donor (*c_D_*(*t*)) and acceptor plates (*c_A_*(*t*)) and analyzed by HPLC. Initial API solutions with a nominal concentration of 100 μM were also measured (*c_D_*(0)). The effective permeability (Equation (1)) and membrane retention (Equation (2)) of the APIs was calculated as follows [19]:(1)$$P_{e}=\frac{-2.303}{A\cdot (t-\tau_{ss})}\cdot \left(\frac{1}{1+r_{v}}\right)\cdot \mathrm{lg}\left[-r_{v}+\left(\frac{1+r_{v}}{1-MR}\right)\cdot \frac{c_{D}\left(t\right)}{c_{D}\left(0\right)}\right]$$
(2)$$MR=1-\frac{c_{D}\left(t\right)}{c_{D}\left(0\right)}-\frac{V_{A}c_{A}(t)}{V_{D}c_{D}(0)}$$
where *A* is the filter area (0.3 cm^2^), *V_D_* and *V_A_* are the volumes in the donor (0.15 cm^3^) and acceptor phase (0.3 cm^3^), respectively, *t* is the incubation time (s), *τ_SS_* is the time to reach a steady state (s), *c_D_*(*t*) is the concentration of the compound in the donor phase at time point *t* (mol/cm^3^), *c_D_*(0) is the concentration of the compound in the donor phase at time point zero (mol/cm^3^), *c_A_*(*t*) is the concentration of the compound in the acceptor phase at time point *t* (mol/cm^3^), and *r_v_* is the aqueous compartment volume ratio (*V_D_*/*V_A_*).

### 2.6. HPLC Analysis

The samples were analyzed using an Agilent 1100 HPLC system equipped with a solvent mixer and quaternary pump, autosampler, column thermostat, and a DAD detector module (Agilent Technologies Inc., Santa Clara, CA, USA). As a stationary phase, a Kinetex C18 column (3 × 30 mm, 2.6 μm) was used; the temperature was kept at 45 °C. Two mobile phase solvents were used in a 3.6 min-long gradient program: water, containing 0.1 *v*/*v*% formic acid (A) and AcN:water 95:5, containing 0.1 *v*/*v* formic acid (B). At the start of the gradient program, the column was flushed with eluent 2% B for 0.3 min and then it reached 100% B within 1.5 min, 100% B was kept for another 0.6 min, and then B dropped to 2%. Chromatograms were collected and processed with ChemStation software (Version B.04.03.); peaks were integrated either at wavelength 220 nm or 254 nm based on spectral characteristics.

## 3. Results and Discussion

### 3.1. Characterization of Investigated Drug Compounds

The identification of the tissue-specific nature of the PAMPA permeability system and the differences in the permeability behavior of natural-lipid-extract-based artificial membranes required the selection of a structurally and physicochemically diverse set of investigated compounds. Accordingly, as a result of the two-level diversity criteria system (see Section 2.2), 32 different molecules were selected and the distribution and structural diversity of their physicochemical parameters from a medicinal chemistry approach are presented in Figure 2 and Figure 3.

The physicochemical diversity is presented in a reduced medchem parameters environment (Figure 2), which clearly shows that the selected APIs cover a relatively diverse range of molecular weights (Mw ≤ 500), lipophilicity (clogP ≤ 5), ionization-dependent lipophilicity (clogD_pH7.4_
≤ 4), and polar surface areas (30 Å^2^< TPSA < 140 Å^2^), as parameters according to Lipinski’s rule of five (Ro5) [20], beyond Ro5 [21] and Veber’s rules [22]. Based on the predicted pK_a_ (See Appendix A) values of the compounds, the net charges at pH 7.4 corresponding to the PAMPA test are also given. The distribution of net charges can be matched to the results of the chemogenomic classification map of the approved small-molecule drugs [23].

In Figure 3, the selected set of investigated APIs in the chemical space of approved drugs (2560) was visualized by a t-SNE plot. The colored points representing the acid/base character of the 32 drugs are equally distributed over the map. Their distance from each other correlates with their structural diversity. The first thing the map shows us is that they fit within the space of known drugs. The second is that they are not clustered within one or a few groups but populate the whole space. To summarize the results of the above classification criteria, in physicochemical and structural terms, it verifies that the selected APIs are a diverse and representative subset of the drug-like chemical space.

### 3.2. Characterization of Natural Lipid Extracts

#### 3.2.1. Chemical and Basic Physicochemical Characterization

The lipid composition of the tissue extracts provided by the manufacturer (Avanti Polar Lipids Inc., Alabaster, AL, USA [24]) can be seen in Figure 4. The most detailed information is available about the liver extracts, which contain the most glycerophospholipid components, while the brain and heart tissues are mostly undefined, with them having 30–60% unknown components. It is worth mentioning that for all the tissues, the main glycerophospholipid components are phosphatidylcholine (PC) and phosphatidylethanolamine (PE). As a result of further extraction, polar lipid extracts are enriched in glycerophospholipids, with them containing fewer neutral lipids (NLs) and unknown components. The most markable difference appears in the comparison of BPLE–BTLE. It is also important to emphasize the (increased) tissue-specific presence of certain phospholipids, such as brain–PS, (PA) (BPLE–PE), heart–Cl, and liver–(PI). CHO.

In connection with these results, the difference in the glycerophospholipid composition of natural lipid extracts as a function of the interactions between the lipid components and the APIs is interesting to investigate. However, it is important to take into account that the solubilized structure of the lipid molecules in the pores of the PAMPA filter membrane can be also influenced by the lipid–solvent ratio in accordance with the findings cited in the Introduction section (Carrara’s [12] and our lipid-driven hypothesis [5] for brain-specific transport). Based on the model proposed by Thomson et al. [13], the lipids can be present in the bulk of the solvent separately (fully solvated form), in a unilamellar form, and a multilamellar form, and they can approach the double lipid monolayer form with decreasing amounts of solvent (Figure 1), given that this final structural arrangement favors not only the natural lipid membrane structure but also the specific transport of separated lipids. Accordingly, we also tried to ensure this lipid arrangement in our tests by using a membrane system with a reduced dodecane content (see the solvent mixture system in Section 2.5) in the PAMPA models. In addition to providing these conditions, the primary task remained to identify the differences in the interaction of glycerophospholipids (Figure 4) and APIs. Since API molecules first contact with lipid molecules at the unstirred boundary layer of the donor aqueous phase and lipid solution, following Avdeef’s tetrad equilibria model [25], the primary interaction takes place between the drug and the polar head of the lipid. In the case of glycerophospholipids, the head groups are positive (choline and ethanolamine) and the zwitterionic (serine), neutral (inositol), and negative charge of phosphoric acid can influence the ionic interaction with the APIs (surface ion-pairing—SIP). At the same time, these head group fragment elements shape the net charge of each lipid (net neutral (±): phosphatidylcholine—PC and phosphatidylethanolamine—PE; net negative charged (-): phosphatidylinositol—PI, lyso-PI, phosphatidic acid—PA, phosphatidylserine—PS, cardiolipin—CL:2-) and also ensure the combined charge cloud of complex (e.g., tissue-specific) lipid membrane systems. Thus, the net charge of the known phospholipid components and the ratio of negative to neutral (zwitterionic) charge can be calculated for the investigated total and polar lipid extracts (Figure 5).

As Figure 5 shows, brain and heart extracts have a higher (-)/(±) ratio, with them having more net negative lipid components (brain—PS (-), heart—CL (2-)). In the meantime, the liver has the lowest (-)/(±) ratio, due to its high net neutral lipid (PC and PE) content. This difference in the lipid membrane charged state can influence the membrane permeability of ionized drugs (Figure 2D).

#### 3.2.2. Zeta Potential

In order to further characterize the tissue-specific lipid extracts, liposomes were made to measure the zeta potential of aqueous liposome suspensions in a pH 7.4 PBS medium (Figure 6).

All of the zeta potentials were between -20 and -40 mV with low deviations, which suggests very stable liposomal suspensions. In the case of each tissue, significant differences were found between polar and total extracts. However different tendencies have been observed for the tissues: while the liposomes from the brain and heart polar extracts showed higher negative zeta potential compared to their total pairs, in the case of the liver, an opposite tendency was found. The increased negative zeta potentials in the relation to the polar-total lipids for the brain and heart extracts can be explained by the higher total glycerophospholipids content and the presence of negatively charged PS (-) and Cl (2-) lipids, respectively (Figure 4). The zeta potential values of liver lipid extracts suggest that they behave differently in the liposome solutions than in the brain and heart extracts (Figure 6); this could be in accordance with the elevated glycerophospholipid ratio and its previously mentioned charge ratio, as well. However, the significant difference in the value of the zeta potential of the liver’s total-polar lipids cannot be explained by either the lipid compositions or the ratio of the net charge of the phospholipids. For easier identification of possible correlations between the permeability of the investigated APIs and the zeta potential values of the lipid extracts, the order of the zeta values is also presented in Figure 6.

### 3.3. In Vitro Tissue-Specific Permeability

The PAMPA permeability test of 32 compounds with diverse physicochemical and structural properties was performed in pH 7.4 PBS medium with a reduced n-dodecane and relatively high lipid concentration (106.7 mg/mL), thereby ensuring uniform lipid-driven drug transport and modeling tissue/interstitial space pH and electrolyte concentration.

When comparing the permeability of the 32 drug molecules on different tissue-specific polar and total extracts, statistically significant differences could be found only between liver total and heart total lipids (Figure 7, *p* < 0.05, one way ANOVA, Tukey’s multiple comparisons test). In order to find further interrelations, we separated the investigated molecules into four charge categories: positive (bases), negative (acids), neutral, and zwitterionic (amphoteric) compounds. The main charged forms at pH 7.4 were calculated based on pK_a_ values, using Marvin Skecth/ChemAxon prediction software v.20.21.0 (Figure 2D). In Figure 8, tissue-specific and total-polar-related permeability is plotted based on the charge state of the APIs.

General trends can be observed, since positive and neutral compounds show higher permeability, while negative and zwitterionic compounds’ penetration is much lower. This can be explained by the Avdeef’s surface ion-pairing (SIP) hypothesis (ionic interaction between the net charge surface of the lipid extract membrane and the ionized API) [22]. The permeability values for liver lipids (especially the total) are the highest for all charge state categories (neutral, positive, negative, and zwitterionic compounds). In general, the permeability of drugs on liver extracts is higher than the mean values. This result can be partly explained by the increased glycerophospholipid concentration of the liver lipid extracts and can also be related to the main pharmacokinetic function of the liver, i.e., the effective metabolic clearance of lipophilic xenobiotics must be accompanied by their increased hepatic uptake with positive logD_pH7.4_ and less structure-specific correlation [26]. An interesting pattern for liver lipids can be observed, as neutral compounds showed higher permeability than positive ones only here regarding the total-polar lipids relation. Furthermore, the highest zwitterionic permeability can be found for LPLE. In contrast to the liver, when using heart and brain lipids in the assay, compounds show lower permeabilities on average. On heart total lipids, the permeability of zwitterionic compounds could not be quantified due to detection limits. Another interesting correlation can be observed regarding BPLE, i.e., here the highest difference in permeability between positively and negatively charged APIs can be identified, which proves the increased relevance of PS in SIP and serves as further evidence for the successful use of BPLE in the BBB-specific permeability assay [5,8,12].

By correlating total and polar permeability values (intra- and inter-tissue relations), the differences between the polar and total extract are more visible (Figure 9). During the examination of the different tissues by linear regression, brain and heart tissues were found relatively similar, as permeability values correlated well for both polar and total extracts (R^2^ > 0.8). In the meantime, when comparing the heart–liver and brain–liver extracts, much lower R^2^ values were found indicating that liver extracts act differently from the other two tissues. The lowest R^2^ value was found for heart total-liver total comparison, confirming the ANOVA test, which recognized the only significant difference in this case.

By correlating polar and total permeability values for certain tissues, the brain extracts were found the most similar (R^2^ = 0.727) while the liver extracts showed the biggest difference (R^2^ = 0.521).

By plotting the abovementioned linear regressions, further patterns can be observed for the different tissues. (Figure 10) In the case of brain polar and total permeability values (Figure 10C), most of the data points are located in an area very close to the 45° line, suggesting that brain polar and total permeability values are quite similar for most of the compounds. Meanwhile, bigger differences can be observed for the other two tissues: for the heart tissue, the data points are clustering below the line, indicating that the heart polar permeability of a compound is usually higher than the total value (Figure 10A). On the contrary, in the case of the liver extracts, the compounds usually penetrate through the liver total extract more easily than through the polar extract (Figure 10B). The liver-associated generally higher permeability can be explained with the higher glycerophospholipid content of these tissues, although the somewhat greater difference between liver total and polar values is presumably not the result of the lipid composition, since they seem to be quite the same according to the manufacturer. However, this difference is also verified by the zeta potential values, which suggests that there should be some structural differences between the two extracts. Summarizing the compound pattern in Figure 10A–C, nitrazepam (**14**) and verapamil (**31**) uniformly showed higher permeability in total lipids, while buspirone (**2**) and dibucaine (**5**) showed higher permeability in polar lipids. It is also interesting to highlight the changes in the total-polar permeability of flutamide (**7**) and levomepromazine (**11**). While the permeability of **7** was higher in the liver total extract, it was higher in the brain polar extract. Similarly, this change in permeability was shown by **11** in the context of heart total-liver polar. Reviewing the basic physicochemical parameters of these compounds showing lipid selective permeability patterns, all of them are positively charged or neutral, logD_pH7.4_ values are between 1.3–3.3, and TPSA values are between 50–85 Å^2^ (except for cpnd **11** = 15.7 Å^2^), which satisfy the rule of medchem thumbs by providing the increased lipid interaction for these APIs.

If we focus on the inter-tissue relationships, regression pairs with R^2^ < 0.7 could be remarkable, because only slight correlations were found. Accordingly, for this aspect, two systems can be found in Figure 11, liver total–brain total (Figure 11A) and liver total–heart total (Figure 11B). The unique permeability pattern of **2**, **5**, **7**, **11**, and **31** compounds can also be seen in these plots.

### 3.4. Cheminformatic Analysis of Tissue-Specific Permeability Data

#### 3.4.1. Principal Component Analysis

Characteristic patterns in the measurement permeability data were also searched by applying principal component analysis (PCA). The result of the PCA analysis is shown in Figure 11. On the biplot, the axes correspond to the two largest principal components (PCs), the arrows represent the assays (different lipid systems) in the subspace of the first two PCs, and the dots represent individual compounds. Both assays and compounds are vectors in the PC analysis. In the case of the compounds, these are vectors pointing from the origin to the dot in the plot. Conventionally, the arrows are omitted in this case for clarity.

Similarity between different assays is measured by the angle (γ) between the corresponding arrows. In addition, the most orthogonal polar-total extract pair belongs to the liver extract (LTLE-LPLE). For example, BPLE and BTLE result in more similar permeability than HPLE and HTLE. The measured activity of a compound (dot) on a given assay (arrow) is proportional to the inner product ($\left|a\right||b|cos\gamma$). Meaning, the logP_e_ value is proportional to the length of both vectors and decreases as the angle between them grows. For example, compounds **2**, **5**, **7,** and **31** have high permeability on HTLE, while **19**, **25**, **26,** and **32** have low (this can also be checked in Figure 11B). In another approach, if the PCA biplot is divided into four spatial parts (A–D), compounds with higher permeability are located in A,C (left), while compounds with lower permeability are located in B,D (right). Furthermore, the compounds located in A,B (upper) have more heart- and brain-specific permeability, while compounds in C,D (lower) part have more liver-specific permeability. This also means that the permeability of the compounds (e.g., compounds **10**, **23,** and **30**) near the abscissa of the PCA biplot is independent of the investigated lipids. Macroscopically, the pattern of Figure 12 dots supports that basic and neutral compounds typically have higher permeability (similar to Figure 8).

#### 3.4.2. Correlation Analysis

The Spearman correlation between the permeability values and several molecular descriptors mainly related to medchem rules was calculated to identify significant indicators of the permeability trend of the specific lipid extract. The experimental dataset consists of the 32 investigated APIs described by six lipid-specific PAMPA permeability values. Note that Spearman correlation depends only on the order of permeabilities, therefore, it is invariant to logarithmic transformation. An extended set of descriptors were used for this experiment: 5 general physchem descriptors for the medchem rules using Marvin Sketch (Chemaxon Ltd.), 7 specific ADME descriptors using ACD/Percepta (ACD/Labs), and a further 19 structure-specific molecular descriptors computed by RDKit. The table containing the descriptor values for all molecules can be found in Appendix A.

The results of the analysis are shown in Figure 13 Red cells indicate a positive correlation, while blue cells indicate a negative correlation. As a macroscopic pattern, several descriptors used in lead compound and drug candidate selection have a significant but non-lipid-specific correlation with permeability values. Thus, positive correlation to logP and logD_pH7.4_ and negative correlations to TPSA, the number of H-bond donors (N_HBD_), and the heteroatoms (N_HAC_) has been assigned. This result is consistent with the extensive use of these parameters as robust descriptors of in silico models for ADME-T processes such as intestinal absorption [27,28], brain distribution [29], metabolism [25], toxicity risk [30], or hERG inhibition [31] with similar correlation signs. Moreover, these descriptors are also generally used in such virtual screening tools as an earlier mentioned Ro5 [16] or other filters (Ghose’s [32], Veber’s [18], Egan’s [33], and Muegge’s [34]) [35], which provides further evidence for the internalization of PAMPA models in early drug discovery.

Non-lipid-specific correlations were found between the permeability values and predicted pharmacokinetic parameters, such as Caco-2 and jejunal permeability, logBB, passive transcellular absorption (positive), solubility (logS_pH7.4_), and paracellular absorption (negative). These findings provide the validity of the PAMPA model as a general permeability model with high-throughput but medium specificity in preclinical in vitro–in vivo translations. In contrast to these findings, positive lipid-specific correlations can be observed for acidic pK_a_ (pK_a,a1_)-liver total extract (LTLE), amide bond count–liver polar (LPLE), and brain lipids (B(P/T)LE) and negative correlations can be observed for aromatic heterocycle count-heart total extracts (HTLE) and H-bond acceptor count (N_HBA_)-liver lipid (L(P/T)LE) and heart lipid (H(P/T)LE) extracts. The observation that pK_a,acidic_ is positively correlated with LTLE-specific permeability is in good agreement with the description of Austin et al. [36], i.e., the increasing acidity of the APIs reduces the affinity to hepatocytes. It is also interesting to identify a positive correlation between LPLE-, B(P/T)LE-specific permeability, and the number of amide functions, which has a critical role in the composition of many classic small-molecule [37], peptide, or peptidomimetic drugs [38]. Here, it is worth emphasizing the observation of Ono et al. in that free amide hydrogens as hydrogen bond donors interact with the membrane; moreover, the water–lipid membrane interface plays an important role in the development of high permeability [39]. Furthermore, in the case of cyclic peptides, it was identified that the side chain amide moiety directly modify the hydrophobicity and affects their permeability and metabolism [35]. However, it is much more complicated to address the negative correlation between heteroaromatic count and HTLE-specific permeability. Because, e.g., referring to the findings of Ritchi et al., it has only a modest detrimental effect on aqueous solubility and also has a modest but beneficial effect on logD_pH7.4_ [40]. The partial tissue-specific function of N_HBA_ cannot be examined independently of the Spearman correlation pattern found in N_HBD_. Similar to Lipinski’s Ro5 (N_HBA_ ≤ 10, N_HBD_ ≤ 5) [16], N_HBA_/N_HBD_ balance suggests an, e.g., better aqueous solubility (N_HBA_
> N_HBD_) [41], BBB permeability (N_HBD_
> N_HBA_) [42], reduction in P-glycoprotein efflux ratio (at lower N_HBD_) [43], and biological target recognition [44], which is related to the HBD acidity and HBA basicity nature [45,46], and that the greater polarity and solvation of drug-like compounds is ensured by the greater presence of HBA on the molecular surface [47]. This HBD–HBA imbalance frustration can also be identified by the data pairs of significant negative correlations for N_HBA_—LTLE-specific and N_HBD_—brain-lipid-specific permeability.

We also derived the differential quantities (“polar lipid preference”) by subtracting the total lipid-specific permeability values of each experimental quantity from the polar lipid-specific values. Spearman correlation was also calculated between the molecule descriptors and the polar lipid preference. Note that, here positive correlation means, the higher the descriptor values, the higher the relative permeability in the polar lipid case versus the total lipid case. Figure 14 shows that while it is not general, we can still find occasions when the difference can significantly be characterized by a freely and easily computable descriptor. In the case of the liver, for example, the correlation with the strongest acidic pK_a,a1_ or logD_pH7.4_ value indicates that increasing acidity or lipophilicity at pH 7.4 will reduce the polar lipid preference of the compound.

Considering our previous observation based on Figure 4 and Figure 6, that the presence of phospholipids increases in the polar lipid extracts in a polar-total comparison, the descriptors belonging to the Spearman correlation values marked in the red color in Figure 14 can also be addressed as descriptive parameters of permeability related to phospholipids. Accordingly, the red parameters of the RDkit can be assigned to polar lipid preference or otherwise to phospholipid-dependent permeability, which are positively correlated with descriptors closely related to molecular size (Sp3 fraction, number of (hetero)atoms and saturated rings, and exact molecular weight) and accessible surface area defined by Labute [48]. However, it is important to note that this result was obtained by examining a drug-like compound in a relatively narrow molecular mass range (see Figure 2A). Among the predicted pharmacokinetic parameters, we found a polar lipid preference only for logBB, which is of great importance for the PAMPA-BBB model, because it serves as further confirmation for the application of the BPLE-based PAMPA model commonly used by our research group and others, in the early phase in vitro BBB permeability studies.

## 4. Conclusions

In this study, we compared the permeability behavior of standardized total and polar isolates of tissue-specific (heart, liver, and brain) lipids on 32 active pharmaceutical agents (APIs) with structural and physicochemical diversity. To identify the relationships between the structure- and lipid-specific permeability, experiments with a robust parallel artificial membrane permeability assay (PAMPA) and in silico analysis of 30 structural and physicochemical descriptors used in medicinal chemistry were applied. Considering the net charge (at pH 7.4) of both the lipid extracts and the investigated APIs, we proved that the average permeability of neutral and positively charged (basic) compounds was increased compared to anionic (acidic) and zwitterionic compounds, which corresponds to the surface ion-pairing (SIP) hypothesis. In the total-polar-lipid extracts, the difference in permeability was clearly distinguishable within the individual tissue isolates. The *polar lipid preference* observed in the case of the brain lipid, which was identified by the significant correlation with the in silico logBB parameter, confirmed the valid use of BPLE in the BBB-permeability screening. We managed to prove the tissue-specific nature of natural lipids with some physicochemical descriptors (such as p*K_a,acidic_*—LTLE, number of amide bonds—LPLE, BPLE/BTLE, number of aromatic heterocycle—HTLE, and the HBA/HBD imbalance pattern for the lipids), which can be a suitable starting point for the mechanistic understanding of heart- and liver-specific permeability processes, which have been less studied so far, and for the development of novel in silico permeability models. Overall, our obtained results confirm that the mechanism of drug permeation in the in vitro lipid-based (cell-free) PAMPA system can be primarily defined by the effect of phospholipids. Thus, on the basis of the PAMPA-BBB (PBLE-based) model previously validated with in vivo pharmacokinetic data (logBB for human), it can be confirmed that polar natural lipid extracts richer in phospholipids are better suited to ensuring the tissue-specificity of in vitro penetration models than total lipid extracts. Although, in contrast to brain lipid isolates, in vivo human pharmacokinetic values corresponding to the logBB parameter are not available for liver and heart tissue drug distribution, our studies clearly show that the drug permeability property measured in liver-specific lipid isolates differs greatly from heart and brain lipids. The higher average permeability values identified for the liver lipid extracts are in good agreement with the physiological function of the liver, that is, the higher proportion of APIs taken up by the hepatocytes helps the increased metabolic clearance and elimination of the drugs. Thus, our studies confirm that polar natural lipid extracts can serve as a suitable basis for predicting not only the brain occupancy but also the tissue distribution of drugs in the liver and heart.

## Figures and Tables

**Figure 1 pharmaceutics-15-00899-f001:**
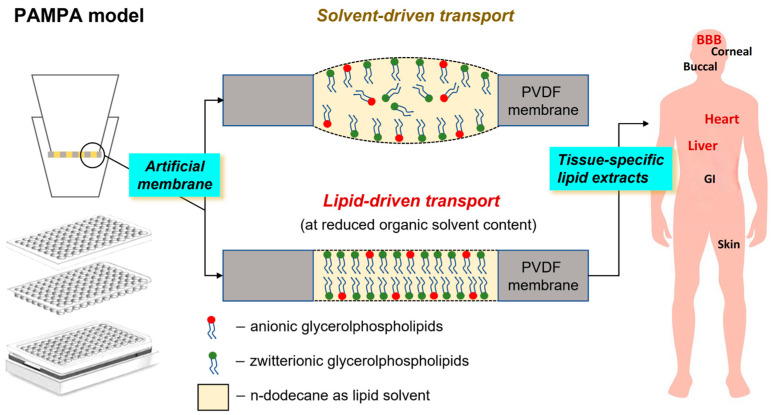
Tissue-specific PAMPA models and *lipid-driven hypothesis* for the in vitro non-cell permeability model (proposed by Thomson et al. [13] and our group [5]). BBB–blood–brain barrier and GI—gastrointestinal tract.

**Figure 2 pharmaceutics-15-00899-f002:**
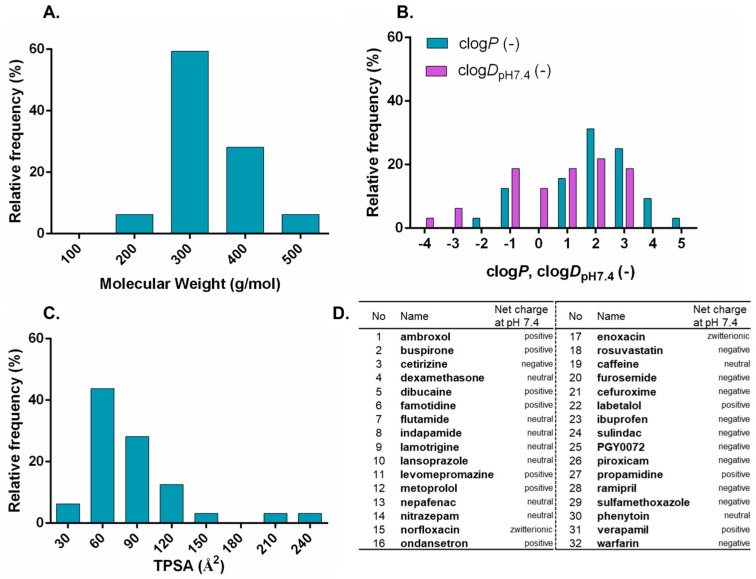
Frequency distributions of the physicochemical properties of the investigated compounds. (**A**) Molecular weight, (**B**) lipophilicity, (**C**) topological polar surface area, and (**D**) net charge at pH 7.4. The “D” section of the figure contains all of the active pharmaceutical ingredients (APIs) investigated along with their code numbers.

**Figure 3 pharmaceutics-15-00899-f003:**
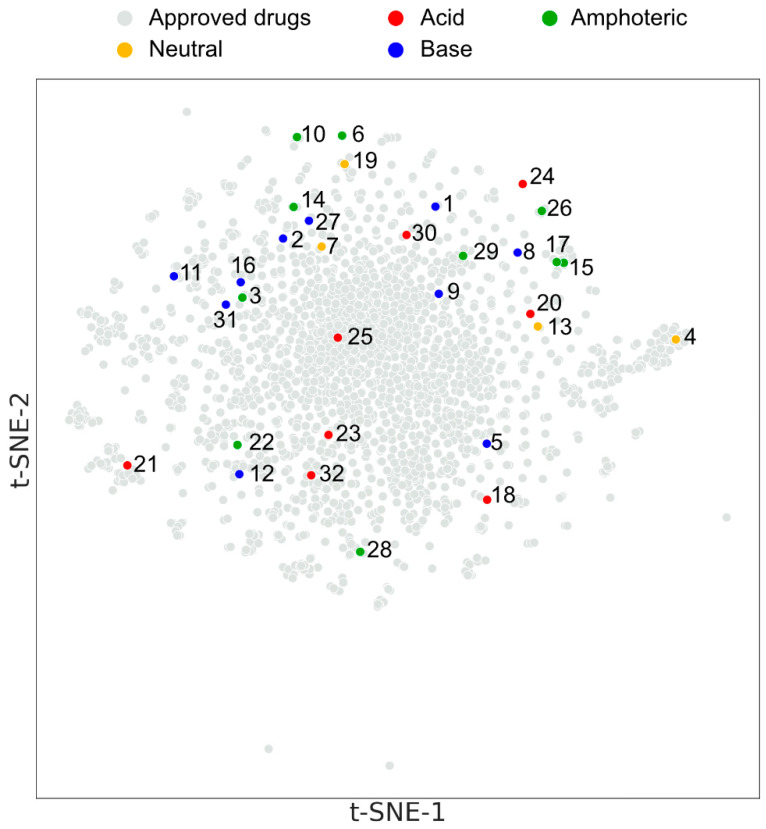
t-SNE plot of the APIs used in the experiment. The database of the ‘approved drugs’ contained 2560 molecules. The compounds are numbered as in Figure 2D. t-SNE—t-distributed stochastic neighbor embedding.

**Figure 4 pharmaceutics-15-00899-f004:**
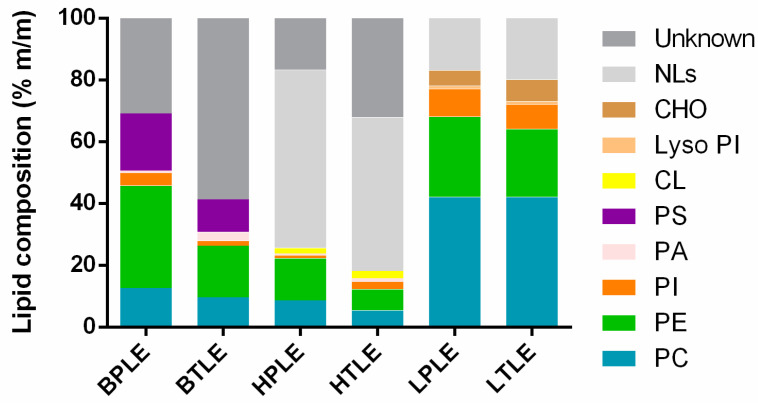
Lipid composition of the tissue-specific extracts. BPLE—brain polar lipid extract, BTLE—brain total lipid extract, HPLE—heart polar lipid extract, HTLE—heart total lipid extract, LPLE—liver polar lipid extract, LTLE—liver total lipid extract, PC—phosphatidylcholine, PE—phosphatidylethanolamine, PI—phosphatidylinositol, PA—phosphatidic acid, PS—phosphatidylserine, CL—cardiolipin, Lyso PI—lyso-phosphatidylinositol, CHO—cholesterol, and NLs—neutral lipids.

**Figure 5 pharmaceutics-15-00899-f005:**
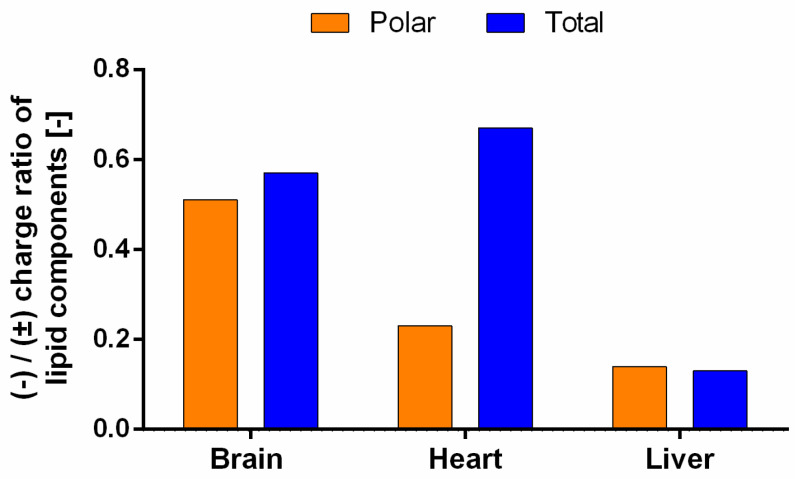
Charge ratio of net negative and net neutral known lipid components in tissue specific lipid extracts at pH 7.4 (Values of charge ratio: BPLE: 0.51, BTLE 0.57:, HPLE: 0.23, HTLE: 0.67, LPLE: 0.14, and LTLE: 0.13).

**Figure 6 pharmaceutics-15-00899-f006:**
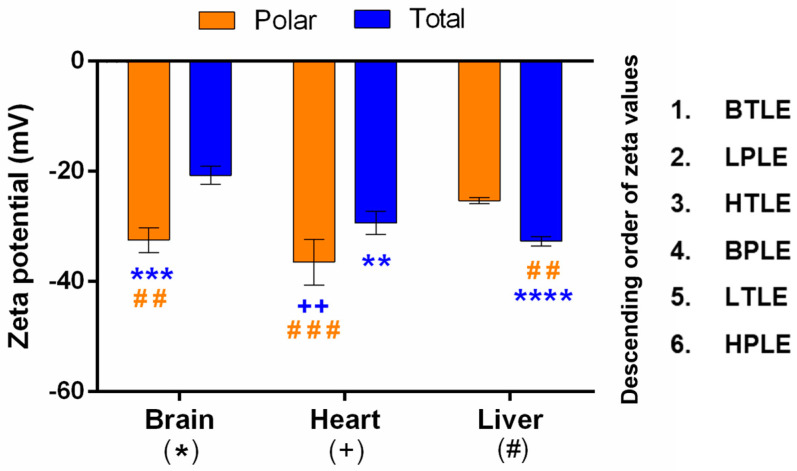
Zeta potential of tissue-specific lipid extracts (in liposome suspensions). All data represent the mean value of zeta potentials with three replicates. Significant differences are marked with symbols respectively: * significant compared to brain lipid, ^+^ significant compared to heart lipid, and ^#^ significant compared to liver lipid. The color of the symbols depicts the comparative polar (orange) or total (blue) extract; **/^++^/^##^ 0.001 < *p* < 0.01, ***/^###^ 0.0001 < *p* < 0.001, **** *p* < 0.0001 (Graphpad Prism; two-way ANOVA, Sidak’s multiple comparisons test), BTLE—brain total lipid extract, LPLE—liver polar lipid extract, HTLE—heart total lipid extract, BPLE—brain polar lipid extract, LTLE—liver total lipid extract, HPLE—heart polar lipid extract.

**Figure 7 pharmaceutics-15-00899-f007:**
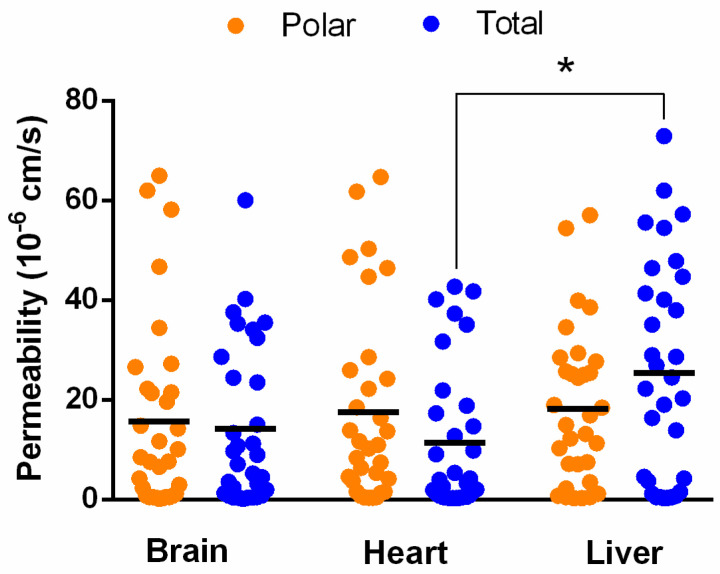
Permeability of 32 drugs on tissue-specific artificial membranes. Scatter plot with a line depicting the mean value with three replicates; significant differences are marked with an asterisk (* *p* < 0.05, one-way ANOVA, Tukey’s multiple comparisons test). For details on the measured data see Appendix A.

**Figure 8 pharmaceutics-15-00899-f008:**
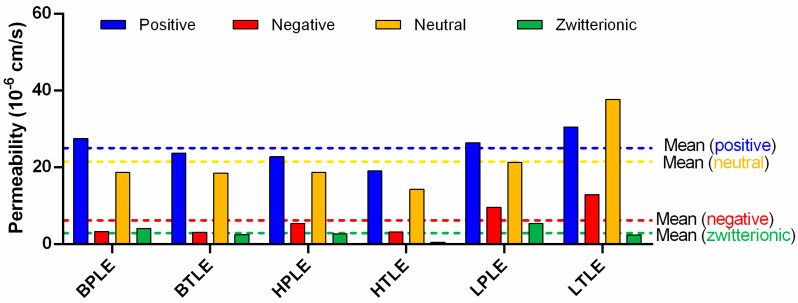
Permeability of 32 drugs on tissue-specific membranes based on charge state at pH 7.4. All permeability data represent the mean of three replicates. Mean values are marked with scattered lines. BTLE—brain total lipid extract, LPLE—liver polar lipid extract, HTLE—heart total lipid extract, BPLE—brain polar lipid extract, LTLE—liver total lipid extract, and HPLE—heart polar lipid extract.

**Figure 9 pharmaceutics-15-00899-f009:**
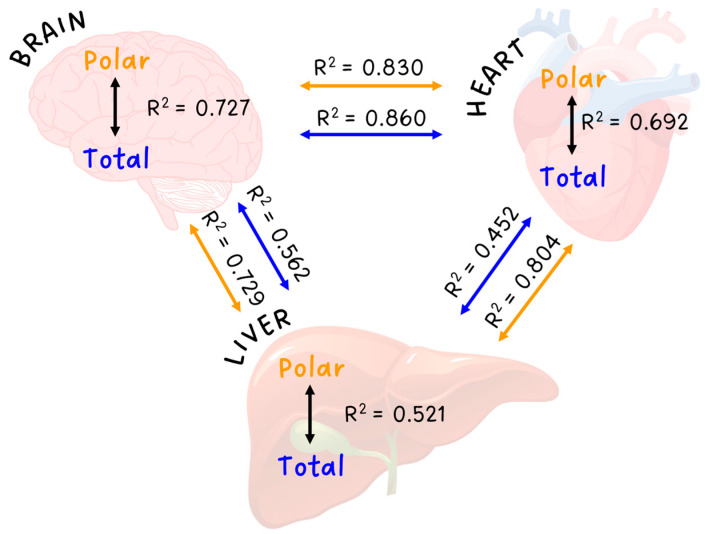
Intra- (total-polar) and inter-tissue correlations based on permeability measurement (n = 3). Determination coefficients are derived from paired linear regressions.

**Figure 10 pharmaceutics-15-00899-f010:**
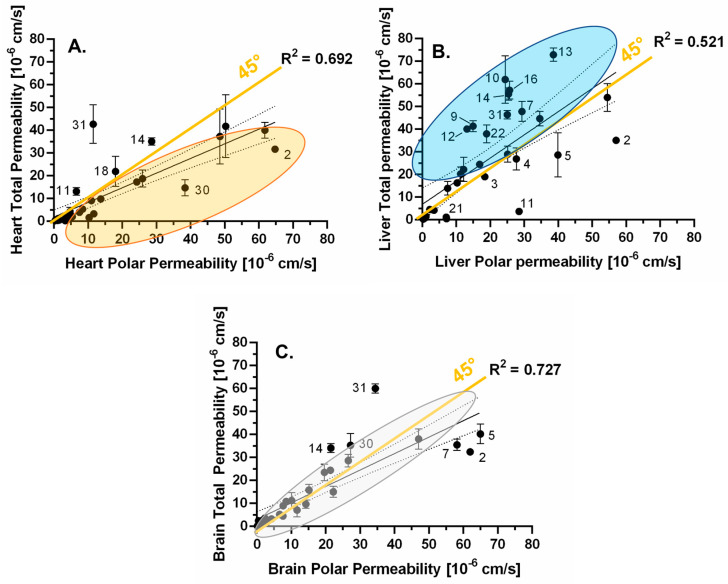
Linear regression between total and polar permeability values (mean, n = 3) for each tissue: (**A**) heart, (**B**) liver, and (**C**) brain. The compounds are numbered as in Figure 2D.

**Figure 11 pharmaceutics-15-00899-f011:**
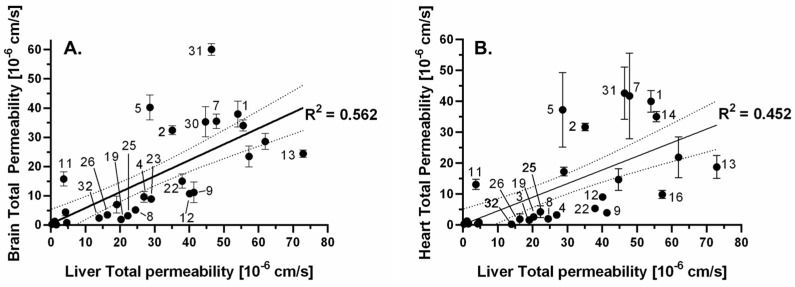
Linear regression between total permeabilities (n = 3), related to liver total permeability: (**A**) brain and (**B**) heart. The compounds are numbered as in Figure 2D.

**Figure 12 pharmaceutics-15-00899-f012:**
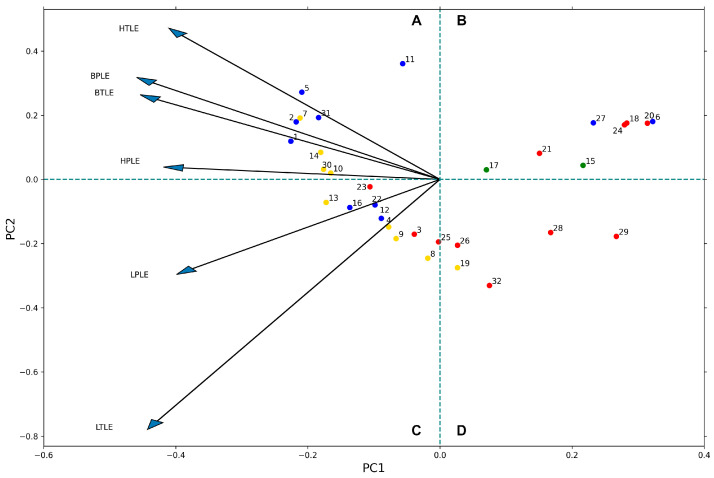
Principal component analysis (PCA) biplot of the permeability data (n = 3). The dots represent compounds and the dots color represent net charge of the compounds at pH 7.4 (blue: positive, red: negative, yellow: neutral, and green: zwitterionic). Arrows represent assays. The compounds are numbered as in Figure 2D.

**Figure 13 pharmaceutics-15-00899-f013:**
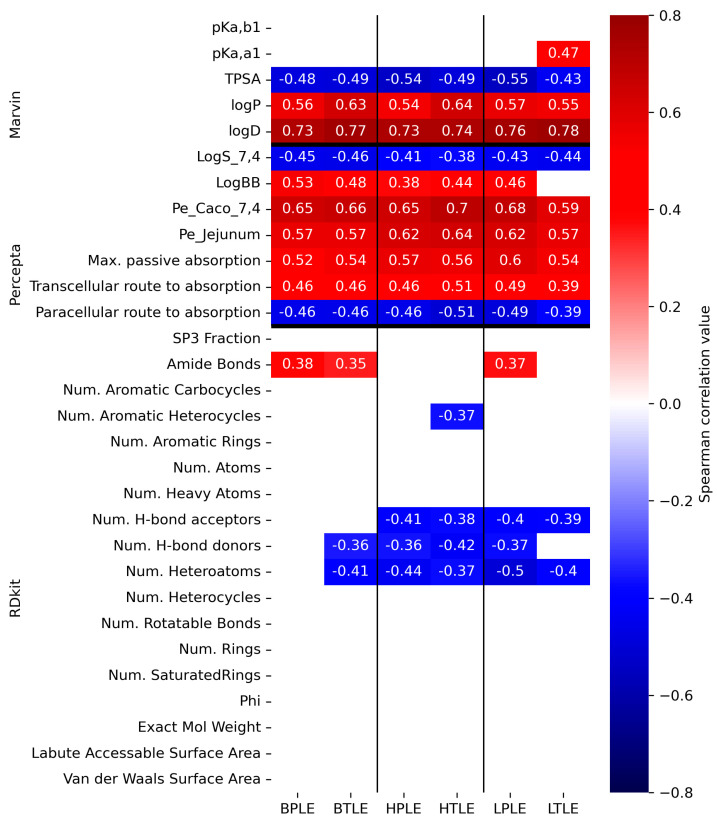
Spearman correlation matrix between the permeability values (n = 3) and the selected descriptors. Only the most significant correlation values are shown (with a significance level of *p* < 0.05). Predicted descriptors and permeability data can be found in Appendix A respectively.

**Figure 14 pharmaceutics-15-00899-f014:**
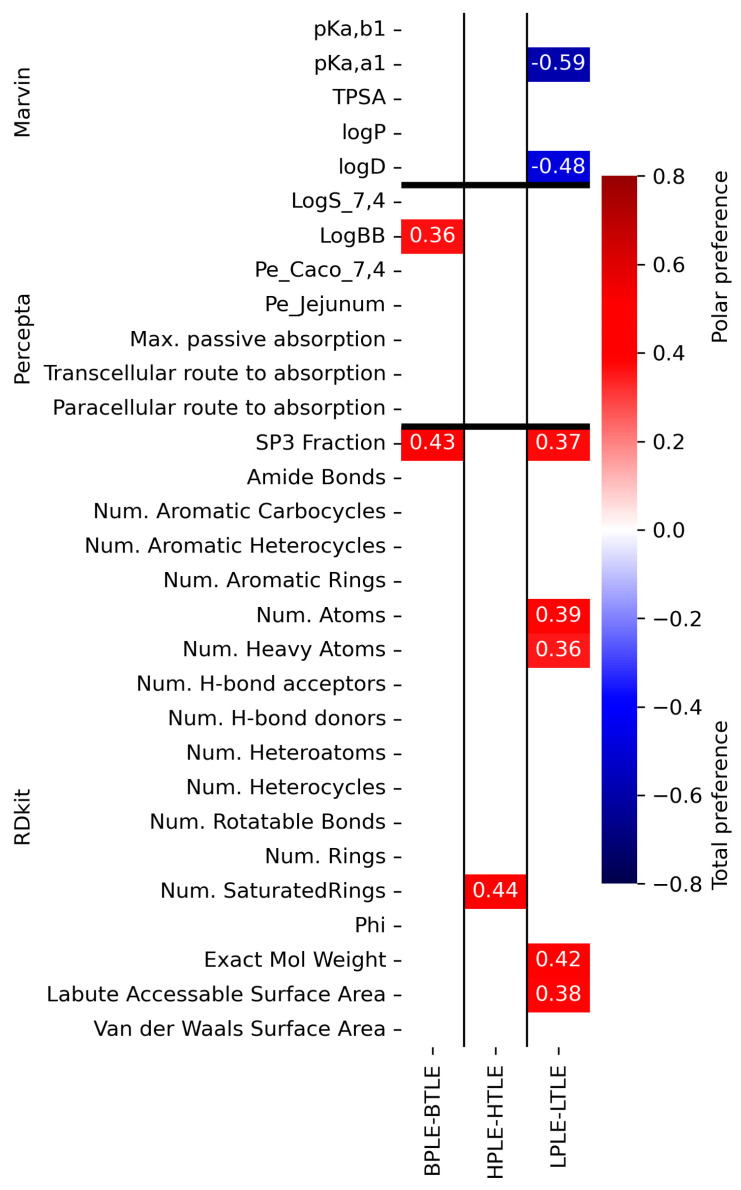
Spearman correlation matrix between the descriptors and the tissue-related polar (red) and total (blue) preferences. Only the most significant correlation values are shown with a significance level of *p* < 0.05. The predicted descriptors and permeability data (n = 3) can be found in Appendix A, respectively.

## Data Availability

All data used in this study are available in the Appendix A.

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
