# Peer review of "Natural Lipid Extracts as an Artificial Membrane for Drug Permeability Assay: In Vitro and In Silico Characterization"

_pharmaceutics, 2023, doi:10.3390/pharmaceutics15030899_

Round 1

Reviewer 1 Report

The manuscript (pharmaceutics-2219241) reported that the drug permeability of natural lipid extracts as an artificial membrane. The study is interesting. The permeability behavior of standardized total and polar isolated of tissue-specific lipids was compared in the text. The physicochemical parameters of the selected compounds were calculated using Marvin Sketh, RDkit, ACD/Percepta softwares. The relationship between the lipid-specific permeabilities and the physicochemical descriptors of the compounds was investigated using linear correlation, Spearman correlation and PCA analysis. The results provide support for understanding the tissue-specific permeability and for developing novel in silico models in the manuscript. Somethings need to be modified before publication. 1) The font of “in vivo” and “in vitro” should be used italics in the text. 2) How much data (n) is used for statistical analysis of the data? The number (n) should be listed under the figures and tables.

Author Response

Reviewer1

The manuscript (pharmaceutics-2219241) reported that the drug permeability of natural lipid extracts as an artificial membrane. The study is interesting. The permeability behavior of standardized total and polar isolated of tissue-specific lipids was compared in the text. The physicochemical parameters of the selected compounds were calculated using Marvin Sketh, RDkit, ACD/Percepta softwares. The relationship between the lipid-specific permeabilities and the physicochemical descriptors of the compounds was investigated using linear correlation, Spearman correlation and PCA analysis. The results provide support for understanding the tissue-specific permeability and for developing novel in silico models in the manuscript. Somethings need to be modified before publication. 1) The font of “in vivo” and “in vitro” should be used italics in the text. 2) How much data (n) is used for statistical analysis of the data? The number (n) should be listed under the figures and tables.

 Reponse:

1. Although, it is an old custom to italicize Latin words, commonly used foreign words such as in vitro, in vivo, ex vivo etc. are not necessary to writ in italic. According to MDPI (https://blog.mdpi.com/2022/06/09/how-to-use-italics/) and also to Springer „Key Style Points”, these words should be formatted upright. (https://resource-cms.springernature.com/springer-cms/rest/v1/content/3322/data/v11). In addition. Several other dictionary and scientific spelling guide consider that commonly used Latin terms should be italicized or not. For example Terms “in vivo”, “in vitro” are not italicized according to Elsevier Guide Style (in accordance to ACS Guide Style)chttps://booksite.elsevier.com/brochures/authors/Text/style.htm. Thus, we would follow this trend.

2. Thank you for the reviewer's comment. In each case, the average values presented in the figures are the results of three parallel measurements. We have replaced the specification of these at all figures (corrections are marked with red words).

Reviewer 2 Report

The work "Natural lipid extracts as an artificial membrane for drug perme-2 ability assay: in vitro and in silico characterization" was reviewed. In general, the work is well presented in form. However, some points should be considered

The authors describe the behavior of 32 drugs in 3 membrane models using the PAMPA methodology and rationalizing the results obtained through physicochemical characteristics of the compounds and membranes. The work presents a large number of results that support in part the hypothesis put forward, since it was evident only with the PAMPA methodology already available by several companies that are responsible for building their membranes for passive diffusion studies such as what was performed in this work, an example of this is PION-INC (https://pion-inc.com/).

Although it is understood that the objective of this work is to know what are the main interactions that drive the passive diffusion of compounds, it is not clear a real conclusion regarding these interactions or what do you postulate as the optimal membrane model according to the type of compound to be studied, or what is the relationship between diffusion in this type of membrane and the biological activity of the compound? Also, not explicitly, what is the relationship of these results with the generation of an in silico model, also considering what is the relationship with cellular models not so complex to use, such as caco-2 cells or more complex three-dimensional models, the latter of great interest to the readers.

Por otro lado, se debe revisar el uso de palabras latinas, estas deben estar en cursiva, algunos ejemplos están en: 

En la línea 82, 85, 119 para decir in vivo .

89, 121, 184 in vitro

101 en silico

Author Response

The work "Natural lipid extracts as an artificial membrane for drug perme-2 ability assay: in vitro and in silico characterization" was reviewed. In general, the work is well presented in form. However, some points should be considered

The authors describe the behavior of 32 drugs in 3 membrane models using the PAMPA methodology and rationalizing the results obtained through physicochemical characteristics of the compounds and membranes. The work presents a large number of results that support in part the hypothesis put forward, since it was evident only with the PAMPA methodology already available by several companies that are responsible for building their membranes for passive diffusion studies such as what was performed in this work, an example of this is PION-INC (https://pion-inc.com/).

Although it is understood that the objective of this work is to know what are the main interactions that drive the passive diffusion of compounds, it is not clear a real conclusion regarding these interactions or what do you postulate as the optimal membrane model according to the type of compound to be studied, or what is the relationship between diffusion in this type of membrane and the biological activity of the compound? Also, not explicitly, what is the relationship of these results with the generation of an in silico model, also considering what is the relationship with cellular models not so complex to use, such as caco-2 cells or more complex three-dimensional models, the latter of great interest to the readers.

Reponse:

Thank you for the reviewer's useful comments. At the same time, the aim of our work was not to identify descriptors for passive diffusion, but rather to identify interaction differences between available natural lipids. For this, we tried to identify correlations and difference descriptors using both macroscopically Fig8.-10. and higher-order statistical methods Fig12.-14., which we hope to describe in detail in the text accompanying the figures. Based on the data, we would feel quite brave to make a conclusion about the correlation between the diffusion and biological activity of the compounds. On the one hand, because the distribution of the active ingredient in each tissue is not yet a causal relationship regarding the effect, on the other hand, the PAMPA model as an early ADME screening tool is unlikely to be suitable for predicting this. The increased tissue-specific permeability identified with PAMPA may rather predict the probability of the presence of the active substance in a given tissue. The correlation with Caco-2 is mentioned in lines 499-504 of the manuscript but given that we identified a close non-specific correlation with the PAMPA models for the number of compounds tested, it can only be concluded that this result validates our system as a general penetration model. We also described this in the section of  Results and discussion.

Por otro lado, se debe revisar el uso de palabras latinas, estas deben estar en cursiva, algunos ejemplos están en: 

En la línea 82, 85, 119 para decir in vivo .

89, 121, 184 in vitro

101 en silico

Response:

Although we could only guess, our response to Rev3's comment is the same as Rev1.

Although, it is an old custom to italicize Latin words, commonly used foreign words such as in vitro, in vivo, ex vivo etc. are not necessary to writ in italic. According to MDPI (https://blog.mdpi.com/2022/06/09/how-to-use-italics/) and also to Springer „Key Style Points”, these words should be formatted upright. (https://resource-cms.springernature.com/springer-cms/rest/v1/content/3322/data/v11). In addition. Several other dictionary and scientific spelling guide consider that commonly used Latin terms should be italicized or not. For example Terms “in vivo”, “in vitro” are not italicized according to Elsevier Guide Style (in accordance to ACS Guide Style)chttps://booksite.elsevier.com/brochures/authors/Text/style.htm. Thus, we would follow this trend.

Reviewer 3 Report

PAMPA is a high-throughput method for predicting drug uptake through biomembranes based on the measurement of passive diffusion of drugs through artificial lipid membranes.  The authors used PAMPA to investigate 31 drug candidates with the potential to cross biomembranes and reach their intended targets in vivo. Understanding the tissue-specific properties of natural lipids is an important step towards developing more effective drug delivery systems and improving our understanding of permeability processes in different tissues. By combining experimental and computational approaches, researchers can gain valuable insights into the physiochemical properties of natural lipids and develop new strategies for drug discovery and delivery. The writing was well written, and the experiments were conducted systematically. However, some modifications are required for publication.

Perform and provide statistically significant values for all experimental results. Details legends are required for all figures.

HCA is a useful analytical technique that can be used to investigate the relationship between a compound's lipid-specific permeability and physicochemical descriptors. It would be nice to derive and present the results through this.

Author Response

PAMPA is a high-throughput method for predicting drug uptake through biomembranes based on the measurement of passive diffusion of drugs through artificial lipid membranes.  The authors used PAMPA to investigate 31 drug candidates with the potential to cross biomembranes and reach their intended targets in vivo. Understanding the tissue-specific properties of natural lipids is an important step towards developing more effective drug delivery systems and improving our understanding of permeability processes in different tissues. By combining experimental and computational approaches, researchers can gain valuable insights into the physiochemical properties of natural lipids and develop new strategies for drug discovery and delivery. The writing was well written, and the experiments were conducted systematically. However, some modifications are required for publication.

Perform and provide statistically significant values for all experimental results. Details legends are required for all figures.

Response:

Re-examining the manuscript in detail, we did not find any part where the statistical significance of the relevant data was not specified. Thus, the significances are given in detail in Fig.6. Fig.7 shows the distribution of the permeability values, there is no point in any statistical evaluation other than the visual presentation. In Fig.8, the compounds were classified based on their acid-base character and the permeability trends are presented accordingly, but in this case only the trends are presented. We do not even write about significant differences in the relevant section, since there are great differences between the compounds due to their internal characteristics, it would be surprising if significance could be identified. In the case of the Spearman correlation (Figs 13-14), the degree of significance was marked in the caption of the figure, and we only provided data in cases where significance was met.

HCA is a useful analytical technique that can be used to investigate the relationship between a compound's lipid-specific permeability and physicochemical descriptors. It would be nice to derive and present the results through this.

Response:

We assume that the reviewer proposed the application of Hierarchical Cluster Analysis under the abbreviation HCA. It is not entirely clear for us how the reviewer suggests to apply this type of analysis in the present situation. Note that we executed measurements on 6 lipid systems, and we used PCA to analyze the latent structure of this 6 dimensional measurement space. HCA requires to define a distance metric over the compounds, which is ambiguous. In our opinion the PCA is a better choice in this situation given the overwhelming majority of the variance (over 95% in this specific case) was explained by the first two principle components, resulting in a highly informative two dimensional embedding.

Round 2

Reviewer 2 Report

<br class="Apple-interchange-newline">The work presented by the authors. corresponds to a study related to the interactions between different membrane models, generated from extracts of polar and total lipids from: liver, heart and brain with biocompounds with different lipophilicity. The authors present the characteristics offered by the manufacturer of the different lipids and the physicochemical characteristics of the different compounds; obtaining a large distribution. Then they evaluated the permeability at pH 7.4 and as expected the interaction was partly controlled by the ionization/polarity of both systems. In addition, the authors correlated the permeabilities observed between the different membrane models used. Although the work presents a large number of experimental results and correlations with values obtained in silico. The conclusions obtained are quite predictable from the formulation of the hypothesis. Therefore, it should focus on the potential applicability of the results obtained for the prediction of the diffusion of biocomposites. Although this can be inferred from the discussions, it should be more explicit to enrich the interest of readers towards an in silico model based on the diffusion mechanism that could exist between the different membrane models used and biocomposites with different nature chemistry.

        The work presented by the authors. corresponds to a study related to the interactions between different membrane models, generated from extracts of polar and total lipids from: liver, heart and brain with biocompounds with different lipophilicity. The authors present the characteristics offered by the manufacturer of the different lipids and the physicochemical characteristics of the different compounds; obtaining a large distribution. Then they evaluated the permeability at pH 7.4 and as expected the interaction was partly controlled by the ionization/polarity of both systems. In addition, the authors correlated the permeabilities observed between the different membrane models used. Although the work presents a large number of experimental results and correlations with values obtained in silico. The conclusions obtained are quite predictable from the formulation of the hypothesis. Therefore, it should focus on the potential applicability of the results obtained for the prediction of the diffusion of biocomposites. Although this can be inferred from the discussions, it should be more explicit to enrich the interest of readers towards an in silico model based on the diffusion mechanism that could exist between the different membrane models used and biocomposites with different nature chemistry.

Author Response

In accordance with the reviewer's request, the aim of our present study was modified in the abstract and introduction. Furthermore, we supplemented the conclusion by clarifying the interpretation of the obtained results.

The corrections have been added to the previous version and marked in red in the uploaded document.